# Optimizing Processing Parameters for NR/EBC Thermoplastic Vulcanizates: A Comprehensive Full Factorial Design of Experiments (DOE) Strategy

**DOI:** 10.3390/polym16141963

**Published:** 2024-07-09

**Authors:** Nataphon Phupewkeaw, Pongdhorn Sae-Oui, Chakrit Sirisinha

**Affiliations:** 1Department of Chemistry, Faculty of Science, Mahidol University, Rama VI Road, Rajdhevee, Bangkok 10400, Thailand; nataphon2611@gmail.com; 2MTEC, National Science and Technology Development Agency, 114 Thailand Science Park, Paholyothin Road, Khlong Nueng, Khlong Luang, Pathumthani 12120, Thailand; pongdhor@mtec.or.th; 3Rubber Technology Research Centre, Faculty of Science, Mahidol University, Salaya Campus, Phutthamonthon IV Road, Salaya, Nakhonpathom 73170, Thailand

**Keywords:** natural rubber, dynamic vulcanization, mechanical properties, morphology, design of experiments (DOE)

## Abstract

This research explores the development of thermoplastic vulcanizate (TPV) blends derived from natural rubber (NR) and ethylene–butene copolymer (EBC) using a specific blend ratio and melt mixing technique. A comprehensive full factorial design of experiments (DOE) methodology is employed to optimize the processing parameters. TPVs are produced through dynamic vulcanization, combining rubber crosslinking and melt blending within a thermoplastic matrix under high temperatures and shear. The physico-mechanical properties of these TPVs are then analyzed. The objective is to enhance their mechanical performance by assessing the influence of blend ratio, mixing temperature, rotor speed, and mixing time on crucial properties, including tensile strength, elongation at break, compression set, tear strength, and hardness. Analysis of variance (ANOVA) identifies the optimal processing conditions that significantly improve material performance. Validation is achieved through atomic force microscopy (AFM), confirming the phase-separated structure and, thus, the success of dynamic vulcanization. Rubber process analyzer (RPA) and dynamic mechanical analyzer (DMA) assessments provide insights into the viscoelastic behavior and dynamic mechanical responses. Deconvolution analysis of temperature-dependent tan δ peaks reveals intricate microstructural interactions influencing the glass transition temperature (T_g_). The optimized TPVs exhibit enhanced stiffness and effective energy dissipation capabilities across a wide temperature range, making them suitable for applications demanding thermal and mechanical load resistance. This study underscores the pivotal role of precise processing control in tailoring the properties of NR/EBC TPVs for specialized industrial uses. It highlights the indispensable contribution of the DOE methodology to TPV optimization, advancing material science and engineering, particularly for industries requiring robust and flexible materials.

## 1. Introduction

Thermoplastic vulcanizates (TPVs) represent an advanced category of thermoplastic elastomers (TPEs) engineered through a technique known as dynamic vulcanization [1,2]. This innovative process merges the advantageous attributes of rubber and thermoplastics, offering the resilience and robustness of vulcanized rubber alongside the processability and moldability of thermoplastics, including capabilities like injection molding [3]. Dynamic vulcanization is characterized by the simultaneous blending and crosslinking of a rubber component within a thermoplastic matrix at elevated temperatures and under significant shear [4,5]. This procedure produces a distinct microstructural composition featuring finely dispersed rubber particles embedded throughout the thermoplastic base. The dynamic vulcanization of the rubber phase also allows the crosslinked rubber to become the dispersed phase, even though the percentage of rubber in the composition is more than 50% (aka. phase inversion).

Dynamic vulcanization, pioneered by Gessler in 1962 and later commercialized by Fisher in 1972 through the creation of the first thermoplastic elastomers [6], marked a pivotal shift in material science by blending crosslinked rubber with thermoplastics to preserve processability. Subsequent refinements by Coran, Das and Patel, and Abdou-Sabet and Fath, using phenolic resins [4,7,8,9], enhanced the rubber-like qualities and flow during processing, leading to the development of TPVs with unique microstructures that enhance their physical and mechanical properties. As a result, TPVs have become integral in various applications ranging from automotive parts to consumer products and medical devices [10,11], due to their adaptability and functional versatility. However, the quest for materials with greater softness and flexibility, particularly in PP/EPDM blends, has highlighted the challenge of balancing softening agents like process oil with the need to maintain mechanical strength, underlining the ongoing need for innovation in TPV composition and processing techniques [12,13].

Thermoplastic natural rubber (TPNR) represents a significant advancement in the family of thermoplastic vulcanizates, drawing attention to its unique characteristics and environmental benefits. TPNR outshines traditional natural rubber (NR) vulcanizates with superior thermal and ozone resistance while providing ease of processing and a broad spectrum of properties at an economical cost [14]. This blend, composed of hard and soft phases by blending thermoplastics with NR, requires melt blending at temperatures above the melting point of thermoplastics, which are higher than a conventional mixing temperature of NR. For HDPE/NR and PP/NR blends [15,16], optimal blending temperatures are around 150 °C and 180 °C, respectively. However, excessively high temperatures can lead to NR degradation, resulting in dark brown blends that are challenging to color correctly, and may require dual pigment systems to achieve the desired appearance, marking the primary drawbacks of PP/NR and HDPE/NR blends.

Polyolefin elastomers (POEs), including ethylene–octene and ethylene–butene copolymers, are gaining recognition as vital additives for enhancing polypropylene (PP) toughness [17,18], offering benefits including high plasticity, aging resistance, and chemical stability at low cost. Ethylene–octene copolymers (EOCs), developed through constrained geometry catalyst technology, are distinguished by their precise molecular weight and comonomer distribution, reducing crystallinity and making them excellent impact modifiers for PP [19,20]. Research on blending EOCs with PP aims to create versatile TPEs and TPVs, examining phase morphology, rheology, and mechanical properties. Babu et al. [21,22] and Yan et al. [23] have explored these blends’ applications, particularly in the automotive sector, while Dey et al. [19] have focused on enhancing TPVs by optimizing processing parameters and vulcanization to improve strength, flexibility, and manufacturability. This includes the targeted distribution of carbon fillers in TPVs made from ethylene–octene copolymer and natural rubber. The aim is to develop TPVs with broad application potential, leveraging DOE methodologies for advanced material integration and manufacturing practices, thus opening new avenues in material science, especially for automotive and industrial applications. However, the potential of POEs in TPNR formulations remains untapped, highlighting a significant area for further research. This unexplored potential indicates an exciting gap for developing TPNRs with enhanced elastomeric properties, capitalizing on POEs’ compatibility with polyolefins and the intrinsic benefits of natural rubber.

This research focuses on improving thermoplastic vulcanizates made from natural rubber (NR) and ethylene–butene copolymer (EBC) blends by fine-tuning crucial processing parameters and dynamic vulcanization methods. The aim is to balance material strength, flexibility, and ease of production. The study uses a design of experiments (DOE) methodology, which involves ANOVA and regression analysis, to improve the integration of the rubber phase into the thermoplastic matrix. This endeavor aims to develop TPVs with wide-ranging applications, and pioneer advanced manufacturing techniques for these innovative materials.

## 2. Materials and Methods

### 2.1. Materials

The primary materials used in this study were natural rubber (NR, STR 5L) and ethylene–butene copolymer (EBC, Exact™ 9371 Plastomer, ExxonMobil Chemical, USA). STR 5L, characterized by its low dirt content and light color, was sourced from Thai Hua Rubber Public Co., Ltd., Thailand. The EBC was provided by Global Connections Co., Ltd., Thailand. The compounding of the NR involved several additives: zinc oxide (ZnO) and stearic acid from Thai-Lysaght Co., Ltd., and Chemmin Co., Ltd., Thailand, respectively; phenolic resin (SP1045) from PI Industry Co., Ltd. (Innovation Group), Thailand; stannous chloride (SnCl_2_) from RCI Lab scan Ltd., Thailand; and Wingstay L from Chemmin Co., Ltd., Thailand. All chemicals were of commercial grade and used as received.

### 2.2. Preparation of TPVs via Dynamic Vulcanization

The production of NR/EBC-based thermoplastic vulcanizates (TPVs) involved a two-stage process utilizing an internal mixer.

The first stage involved preparing the natural rubber (NR) compound on a two-roll mill at 50 °C. NR was masticated and compounded during this stage with ingredients including stearic acid, ZnO, phenolic resin, stannous chloride (SnCl_2_), and Wingstay L, as detailed in Table 1, with the mixing sequence as illustrated in Table 2. The crosslinking system for the dynamic vulcanization process utilizes phenolic resin and SnCl_2_ to selectively enable the vulcanization of the NR phase, while Wingstay L provides resistance to thermal degradation.

In the second stage, this premixed NR compound gained from the first mixing stage was dynamically vulcanized with ethylene–butene copolymer (EBC) in an internal mixer under varied mixing conditions. The composition of NR/EBC TPVs with different blend ratios is detailed in Table 3. Initially, the premixed NR compound and EBC were melt-blended. The NR phase in the blends was then dynamically vulcanized under various processing conditions at a given fill factor of 0.75. The mixing parameters investigated included blend ratio, mixing temperature, rotor speed, and mixing time, as illustrated in Table 4. The resulting TPVs were sheeted using two-roll mills. Test specimens of the blend were compression molded at 120 °C for 5 min. This optimized two-stage approach, which efficiently combined compounding and dynamic vulcanization, led to the development of NR/EBC TPVs with tailored morphology and properties as our innovative approach.

## 3. Mechanical Testing, Rheological and Dynamic Properties, and Morphology Analysis

In this study, a comprehensive suite of tests was conducted to evaluate the mechanical, rheological, and dynamic properties of the thermoplastic vulcanizates (TPVs), along with their morphology.

### 3.1. Tensile and Tear Properties

The tensile properties of the TPVs were evaluated using a universal testing machine (Instron 5566, USA) following ISO 37 (Type I) with a crosshead speed of 500 mm/min and a load cell of 1 kN. This method provided crucial insights into the strength and elongation capabilities of the TPVs.

In addition to tensile testing, the tear properties were assessed as per ISO 34-1 with a crosshead speed of 500 mm/min and a load cell of 1 kN. Test pieces were prepared in the Type B crescent shape. The resistance of the TPVs to tearing under stress could be determined.

### 3.2. Hardness (Shore A)

The hardness of the TPVs, a key indicator of material rigidity and resistance to surface deformation, was measured on specimens approximately 6 mm thick. This measurement was conducted using a Shore A durometer (Wallace H17A, UK), following ISO 48-4.

### 3.3. Compression Set

To evaluate the permanent deformation of the TPVs when subjected to a given compressive load for a specified time, a compression set test was conducted in accordance with ISO 815-1. This test involved cylindrical specimens with a diameter of 13 mm and a thickness of 6 mm, compressed to 25% of their original height for 72 h at room temperature, followed by a relaxation time of 30 min. The compression set percentage was calculated using the following, Equation (1):(1)$$\%CS=\left(\frac{t_{0}-t_{i}}{t_{0}-t_{n}}\right)\times 100$$
where *t*_0_ is the original specimen thickness (mm), *t_i_* is the specimen thickness after testing (mm), and *t_n_* is the spacer thickness (mm). This test is crucial for understanding the material’s ability to return to its original thickness after being subjected to compressive forces, an important characteristic in many practical applications.

### 3.4. Overall Crosslink Density

A swelling test was conducted to determine the crosslink density of the NR phase in the TPVs. Small rectangular specimens, with dimensions of approximately 10 × 10 × 2 mm^3^, were immersed in 50 mL of cyclohexane, a solvent effective for both NR and EBC. This test was carried out at room temperature for 48 h to achieve equilibrium swelling conditions.

Upon completion of the swelling period, the specimens were carefully removed from the solvent. Excess solvent on the surface was absorbed using tissue paper, and the swollen specimens were immediately weighed. The specimens were then dried at 60 °C for 24 h to ascertain their dry weight. The overall crosslink density, an indicator of the network structure within the TPVs, was calculated using the Flory–Rehner equation [24], as shown in Equation (2). This calculation was adjusted to account for both the NR and EBC phases, with a correction for the portion of EBC extracted as an amorphous phase.
(2)$$\left(\nu +EBC\right)=-\frac{1}{V_{s}}\times \left(ln\left(1-V_{r}\right)+V_{r}+\chi V_{r}^{2}\right)/\left(V_{r}^{1/3}-0.5V_{r}\right)$$

In this equation, *ν* represents the number of moles of effectively elastic chains per unit volume of NR (crosslink density), (*ν* + EBC) is the overall crosslink density of the NR phase in the presence of EBC, *Vs* is the molar volume of cyclohexane (84.16 cm^3^/mol), *χ* is the polymer-swelling agent interaction parameter (taken as 0.310 for the interaction between NR and cyclohexane at 25 °C), and *V_r_* is the volume fraction of rubber in the swollen network. The volume fraction of rubber, *V_r_*, is further defined by Equation (3):(3)$$V_{r}=\frac{1}{A_{r}+1}$$
where *A_r_* is the ratio of the volume of absorbed cyclohexane to that of NR after swelling. This approach provides a comprehensive understanding of the network structure within the TPVs, which is crucial for interpreting their mechanical and physical properties [25].

### 3.5. Rheological Properties (RPA)

The stress relaxation behavior of the NR/EBC TPVs was evaluated using a rubber process analyzer (RPA2000, Alpha Technologies, USA). Approximately 5 g of the sample was cut from the milled sheet of NR/EBC TPVs and pre-heated in an oven at 120 °C for 3 min to attain a molten state. Subsequently, the sample was positioned on the lower die of the RPA2000 and allowed to melt further at 120 °C for 1 min inside the test chamber. The test cavity was cooled to 40 °C, and the stress relaxation test was initiated. During the test, the sample was subjected to a strain of 70% and an oscillatory frequency of 0.5 Hz (3.14 rad/s) for 2 min, allowing the measurement of stress relaxation over time under these controlled deformation conditions.

### 3.6. Dynamic Mechanical Properties (DMA)

The dynamic mechanical properties of the TPVs were analyzed using a dynamic mechanical analyzer (Gabo, Explexor^TM^ 25N, NETZSCH-Gerätebau GmbH, Germany) in tension mode with the applied static and dynamic strains of 1% and 0.3%, respectively. A temperature sweep test covered a range from −80 °C to 100 °C at a heating rate of 2 °C/min, maintaining a constant test frequency of 10 Hz (62.8 rad/s). This approach provided insights into the TPVs’ behavior under varied temperatures and dynamic conditions.

### 3.7. Morphology Analysis

The phase morphology of the TPVs was examined with a Nanoscope IIIa peak force tapping atomic force microscope (PF-AFM, Bruker, Germany). Before this observation, the samples underwent preparation with a cryo-ultramicrotome (Leica, Germany) at −100 °C, which is critical for achieving the necessary sample temperature for precision cutting. The size and size distribution of the rubber particles within the TPVs were accurately quantified using Image-Pro Plus 6.5 software, facilitating detailed analysis.

## 4. Experimental Design

The design of experiments (DOE) methodology was applied to methodically explore the impact of various processing parameters on the phase morphology and mechanical properties of NR/EBC TPVs. By adopting a full factorial design strategy, the DOE aimed to scrutinize and pinpoint the processing parameters most significantly influencing the TPV characteristics. This thorough experimental framework was based on a general full factorial model [26], applying all levels of each factor combined with all other factors. This approach guarantees the consideration of every potential combination of factor levels, thus improving the study resolution and mitigating any confounding effects. The primary factors and their specific levels under examination are detailed in Table 4, ensuring a comprehensive analysis.

In this study, Table 5 delineates the levels and interactions of the variables analyzed for their impact on the TPVs’ characteristics, guided by preliminary experiments. Processing temperatures (80 and 120 °C) were chosen to optimize the EBC’s melting without degrading the NR, and rotor speeds (80 and 100 rpm) were selected to ensure thorough mixing and particle breakup. Mixing durations (20 and 25 min) were set to allow complete dynamic vulcanization and prevent the re-agglomeration of NR particles. A total of 32 experimental runs were designed; however, experiments were duplicated and randomized to ensure accuracy and minimize variability, resulting in a final experimental design totaling 64 trials across all factor-level combinations, as tabulated in Table A1 in Appendix A. Minitab 21 for design and analysis, statistical plots, and the F-test ANOVA at a 95% confidence level were employed to assess the influence of processing parameters on the TPVs, ensuring the study’s reliability and the statistical significance of the findings.

The results from the factorial design were succinctly expressed in terms of a regression model. The predictive response of the model was represented by the following first-degree polynomial, Equation (4):(4)$$y=\beta_{0}+\sum_{i=1}^{k}\beta_{i}x_{i}+\sum_{i=1}^{k}\sum_{j=1}^{k}\beta_{ij}x_{i}x_{j}+\varepsilon$$

In this model, *y* represents the predicted responses, with *x_i_* and *x_j_* denoting the coded variables. The term *β*_0_ stands for the global mean, while *β_i_*, *β_j_*, and *β_ij_* are the regression coefficients that correspond to the main effects of the factors and their interactions, respectively. The letter *k* indicates the number of factors involved in the study, and *ε* symbolizes the error term used to quantify the deviation of predicted values from the actual observations defined by the model [26,27,28].

## 5. Results

Part I: Statistical Analysis of Processing Parameters’ Impact on Mechanical Properties of NR/EBC-Based TPVs.

### 5.1. Analysis of Variance (ANOVA)

#### 5.1.1. Tensile Properties of NR/EBC TPVs

Table 6 shows the degree of freedom (DF), sum of squares (SS), mean square (MS), *F*-value, and *p*-value. A higher *F*-value of each factor indicates a more significant influence of that factor on the response. All factors significantly affect mean tensile strength, according to the *F*-values and *p*-values obtained from the ANOVA table. The ANOVA analysis describes how various processing parameters affect the tensile strength (TS) of NR/EBC TPVs, singling out the NR/EBC blend ratio (A) as the key influencer. A balanced 50/50 blend optimally enhances TS by marrying elasticity and plasticity.

Additionally, the analysis recognizes mixing temperature (B) and rotor speed (C) as crucial factors for the quality of mixing and particle dispersion, which in turn impact the vulcanization process and TS. The critical role of mixing time (D) is also highlighted, underscoring the necessity of thorough blending for uniform material distribution and peak tensile properties. The absence of significant interactions among these factors simplifies optimization, permitting isolated adjustments to each. The model boasts considerable explanatory power, as indicated by a high R^2^ value of 86.47% and an adjusted R^2^ of 81.05%, demonstrating an excellent model fit and its capacity to precisely capture the factors affecting TS. Moreover, with a predictive R^2^ value of 72.63%, the model shows a reliable predictive capability for TS in novel samples, further validated by a non-significant lack-of-fit value, reinforcing the model’s accuracy and dependability.

As illustrated in Table 7, the ANOVA analysis sheds light on the elongation at break of NR/EBC TPVs, emphasizing the NR/EBC blend ratio (A) as a crucial factor influencing this property. Elongation at break, indicating ductility, reflects the material’s capacity to stretch before failing, which is vital for applications requiring high flexibility and strength. This research has identified the importance of the blend ratio in enhancing the ductility of TPVs, particularly the proportion of the more elastic NR component. Within the range studied, other factors, i.e., mixing temperature (B), rotor speed (C), and mixing time (D), do not significantly affect this property directly.

While the additional factors do not directly impact the manufacturing process, their significance and potential indirect effects on other material properties are undeniable. The analysis, supported by a high R^2^ value of 93.08%, an adjusted R^2^ value of 90.32%, and a predictive R^2^ value of 86.01%, underscores the critical influence of the blend ratio on elongation at break, demonstrating a strong model fit. The absence of significant interactions between factors simplifies the optimization of the blend ratio, offering a clear route to tailor this property according to specific requirements. This streamlined approach to optimization highlights the manageable relationship between blend ratio and elongation at break, facilitating targeted adjustments to achieve optimal material characteristics.

The ANOVA analyses in Table 6 and Table 7 illuminate the crucial influence of the NR/EBC blend ratio on the tensile strength and ductility of NR/EBC thermoplastic vulcanizates, pinpointing it as the key factor for optimizing these essential mechanical properties. A balanced 50/50 blend ratio proves ideal for tensile strength. In contrast, ductility benefits from a higher proportion of natural rubber, highlighting the pivotal role of the blend ratio without significant direct impacts from other processing parameters, i.e., mixing temperature, rotor speed, and mixing time. This suggests a linear and manageable relationship between blend ratio and the mechanical outcomes of tensile strength and elongation at break, facilitating targeted material property enhancements. The model, with high R^2^ values and the absence of a significant lack of fit, underscores its reliability in predicting these properties, emphasizing the need for focused blend ratio adjustments and further exploration of processing parameters to tailor TPVs for specific industrial applications.

#### 5.1.2. Compression Set of NR/EBC TPVs

The ANOVA analysis in Table 8 decisively underscores the significance of processing parameters on the compression set of NR/EBC TPVs, particularly highlighting the paramount influence of the NR/EBC blend ratio (A) and, to a lesser extent, the mixing temperature (B) on the material’s resilience against permanent deformation. The analysis, marked by a substantial overall *p*-value and a notably high R^2^ value of 96.49%, attests to the model’s accuracy and the blend ratio’s critical role, further complicated by the significant interaction between rotor speed and mixing time. These insights illuminate the dominant factors affecting compression set and suggest a multifaceted optimization approach for processing parameters. This foundational understanding paves the way for future targeted research aiming to develop TPVs with optimized properties to meet the specific demands of varied applications, enhancing their functional suitability and performance.

#### 5.1.3. Tear Strength of NR/EBC TPVs

The ANOVA findings in Table 9 provide a comprehensive analysis of how processing parameters influence the tear strength of NR/EBC TPVs, highlighting the critical importance of the NR/EBC blend ratio (A) for optimizing tear resistance. The significant impacts of mixing temperature (B) and rotor speed (C) on material uniformity and the crucial role of mixing time (D) for achieving a homogeneous blend are vital for enhancing tear strength. While the absence of significant interactions among factors simplifies the optimization process, allowing for targeted parameter adjustments, the model, with moderate explanatory power (R^2^ = 57.77%) and low predictiveness (predictive R^2^ = 14.59%), points to the presence of other unexplored variables that could affect tear strength. This suggests a need for further investigation beyond the current model to identify and incorporate these additional factors, underlining the need for an ongoing quest for improved TPV formulations and processing techniques to respond to diverse application demands with superior tear resistance.

#### 5.1.4. Hardness (Shore A) of NR/EBC TPVs

The ANOVA analysis in Table 10 delivers essential insights into the hardness optimization of NR/EBC TPVs, identifying the NR/EBC blend ratio (A) as the critical factor affecting material hardness. This discovery underscores the necessity of meticulous blend ratio management to satisfy specific hardness requirements, highlighting its direct influence on the mechanical properties of TPVs. Additionally, the analysis reveals that mixing temperature (B) has a negligible impact within the studied range, thus granting manufacturers flexibility in temperature settings without adversely affecting the material’s hardness.

The speed of the rotor (C) has a slight effect on the hardness of the material. This implies that it plays a complex role, requiring further investigation to fully understand its impact on material characteristics. On the other hand, the mixing time (D) significantly affects the hardness, although to a lesser extent than the blend ratio. This highlights the importance of sufficient mixing time for producing a uniform mixture, as it ultimately affects the hardness. With its high R^2^ value of 98.33%, the model demonstrates substantial robustness and predictive accuracy across the examined parameter spectrum. The lack of significant factor interactions simplifies the optimization endeavor, facilitating the targeted adjustment of individual parameters to improve material hardness effectively.

### 5.2. Effect of Mixing Parameters on Mechanical Properties of TPVs

#### 5.2.1. Tensile Properties

The main effects plot analysis for tensile strength, as presented in Figure 1, is crucial for enhancing the tensile strength of NR/EBC TPVs. It highlights the significant impact of specific processing parameters on the material characteristics. The investigation identifies the critical NR/EBC blend ratio (A), with a balanced 50/50 ratio emerging as ideal for optimizing tensile strength. This optimum underscores the importance of harmonizing NR’s elasticity with EBC’s structural support. Any deviation from this blend, especially an increased NR proportion, detrimentally affects tensile strength due to diminished reinforcement by EBC.

Furthermore, the analysis highlights the significant impacts of mixing temperature (B), rotor speed (C), and mixing time (D) on tensile strength. Elevated mixing temperatures might negatively influence tensile strength, so we advocate for lower temperatures to avoid material degradation and promote adequate curing. Additionally, higher rotor speeds and prolonged mixing times reduce tensile strength, possibly due to the mechanical breakdown of rubber particles and excessive heat build-up.

These insights provide actionable guidance, stressing the need for precise control over blend ratios, mixing temperatures, rotor speeds, and mixing durations. By adhering to the identified optimal conditions, i.e., a 50/50 NR/EBC blend, mixing temperatures near 80 °C, rotor speeds at 80 rpm, and mixing times of 20 min, manufacturers have the potential to improve TPV tensile strength markedly.

The main effects plot for elongation at break, illustrated in Figure 2, visually corroborates the ANOVA findings, emphasizing the critical role of the NR/EBC blend ratio in the ductility of NR/EBC thermoplastic vulcanizates. This visual analysis reveals that an increase in natural rubber content significantly enhances the material elongation at break, highlighting the superior flexibility of natural rubber compared to its synthetic counterparts. Conversely, the plot indicates that changes in mixing temperature, rotor speed, and mixing duration have minimal impacts on elongation capacity, suggesting that these factors are less crucial within the tested parameters.

This analysis provides essential guidance for material property optimization, identifying the blend ratio as the critical factor in improving TPVs’ ductility for specific uses. TPVs’ insensitivity in terms of elongation at break to variations in mixing conditions suggests a degree of operational flexibility, which is beneficial in terms of accommodating the natural variability of production settings. This flexibility is precious for maintaining consistent quality despite minor processing condition fluctuations, and underscores the blend ratio’s importance in achieving the desired material performance and ductility.

In conclusion, the main effects plots for tensile strength and elongation at break provide crucial insights into NR/EBC TPV optimization, emphasizing the NR/EBC blend ratio as essential for maximizing both tensile strength and ductility. The optimal 50/50 blend ratio for tensile strength demonstrates the need to balance NR’s elasticity with EBC’s structural support, while an increased NR content notably enhances elongation at break, showcasing natural rubber’s flexibility. These findings suggest precise control over the blend ratio, alongside moderated mixing temperatures, rotor speeds, and durations, as critical strategies for improving TPVs. Due to the minimal impacts of temperature, speed, and time on elongation, operational flexibility in processing allows for quality maintenance despite production variances. Ultimately, this comprehensive analysis underlines the blend ratio’s critical role in enhancing TPV properties, guiding targeted improvements for high-performance material applications.

#### 5.2.2. Compression Set

The main effects plot depicted in Figure 3 crucially elucidates the influence of mixing parameters on the compression set of NR/EBC TPVs, spotlighting the NR/EBC blend ratio (A) as the most influential factor. This analysis reveals that increasing the NR content from a 30/70 to a 60/40 ratio substantially improves the material’s resistance to permanent deformation after compression, identifying an optimal NR content range between 30/70 and 50/50 blend ratios for minimizing compression set.

While mixing temperature (B) does affect the compression set, its impact is considerably less significant than that of the blend ratio, rendering temperature control a secondary concern in minimizing the compression set. Rotor speed (C) and mixing time (D) have minimal effects within the studied ranges, indicating their marginal roles in influencing material resilience post-compression.

This study emphasizes that the blend ratio of NR/EBC is the most critical factor in controlling compression sets. The study highlights that material recovery capabilities are improved with increased NR content due to its elastic properties. While the mixing temperature does play a role, it is overshadowed by the predominant influence of the blend ratio. The limited impacts of rotor speed and mixing time suggest a degree of processing adaptability that does not compromise this key material attribute.

Fine-tuning the NR/EBC blend ratio is the primary consideration for optimal compression set characteristics in TPVs. Despite the lesser significance of mixing temperature, rotor speed, and mixing time on compression sets, their potential effects on other material aspects warrant their inclusion in a holistic processing strategy. This analysis provides a pivotal foundation for targeted material optimization, facilitating the development of TPVs with enhanced performance and resilience tailored to specific application demands.

The interaction effect plot between rotor speed and mixing time (CD), as depicted in Figure 4, offers a detailed examination of their combined impact on the compression set of NR/EBC TPVs. This plot uncovers a complex, interdependent relationship, showing that the optimal rotor speed for minimizing compression set shifts based on the duration of mixing. Increasing rotor speed at a 20 min mixing interval contributes to a lower compression set, whereas at 25 min, a higher rotor speed results in an increased compression set. This observation highlights that the rotor speed’s influence on the compression set is intricately linked to mixing time, a relationship graphically represented by the plot’s diverging lines.

This finding is crucial for refining the manufacturing process. Considering the interaction effects, it implies that a strategic balance between rotor speed and mixing time is essential for optimizing the compression set. Specifically, enhancing rotor speed during shorter mixing periods can improve material dispersion and reduce the compression set, while during more prolonged periods, reducing rotor speed is recommended to prevent overheating or material degradation, which could negatively affect the compression set. This nuanced understanding facilitates a more targeted approach to processing parameters, aiming for optimal material resilience against permanent deformation.

#### 5.2.3. Tear Strength

The main effects plot depicted in Figure 5 is vital in improving the tear strength of NR/EBC TPVs by offering a detailed insight into the impacts of various processing parameters on material durability. It emphasizes the NR/EBC blend ratio (A) as a critical factor, with a 40/60 ratio pinpointed as optimal for maximizing tear strength, thereby illustrating the positive synergy between NR’s inherent elasticity and EBC’s structural support to bolster tear resistance.

Furthermore, the analysis underscores the detrimental effects on the tear strength of certain processing conditions, such as high mixing temperatures, increased rotor speeds, and lengthy mixing durations. Recommendations to counteract these include maintaining mixing temperatures around 80 °C to avert thermal degradation, capping rotor speeds at 80 rpm to lessen shear stress, and limiting mixing durations to 20 min to prevent undue heat build-up and material compromise.

#### 5.2.4. Hardness

The main effects plot for the hardness of NR/EBC TPVs, depicted in Figure 6, offers crucial insights into material hardness optimization via strategic processing adjustments. A key discovery is the critical influence of the NR/EBC blend ratio (A), with an increased proportion of EBC leading to higher hardness levels, especially notable at a 30/70 NR/EBC blend ratio. This underscores the blend ratio as the primary factor in hardness control, requiring precise adjustment for achieving specific material properties.

On the other hand, mixing temperature (B) minimizes hardness, providing manufacturers with flexibility in temperature management without compromising hardness. Rotor speed (C) has a marginal effect on hardness, further emphasizing its minor role. Meanwhile, mixing time (D), despite its statistically significant impact, ranks below the blend ratio in terms of its influence on hardness optimization, suggesting that it plays a supportive role.

These findings highlight the NR/EBC blend ratio as the critical variable for hardness adjustment. Mixing temperature and rotor speed offer operational flexibility, and mixing time presents an additional, though less critical, parameter for fine-tuning. This analysis directs focus toward the blend ratio for material specification achievement while acknowledging other factors’ roles in the broader context of processing optimization.

### 5.3. The Predictive Models for Mechanical Properties of TPVs

The predictive model for NR/EBC TPVs, presented in Table 11, marks a significant milestone in developing and optimizing thermoplastic vulcanizates. It highlights crucial processing parameters, namely, NR/EBC blend ratio (A), mixing temperature (B), rotor speed (C), and mixing time (D). This model provides a precise roadmap for enhancing TPVs’ mechanical properties, including tensile strength, elongation, compression set, tear strength, and hardness. It particularly emphasizes the critical role of the NR/EBC blend ratio, suggesting that finding the right balance between NR’s elasticity and EBC’s structural support is essential for tailoring TPVs to specific application needs.

Moreover, the model regression analysis illuminates the profound impact of the blend ratio on mechanical properties, supported by strong R^2^ values that demonstrate the model’s accuracy in material behavior prediction. The tear strength model, however, with its relatively lower R^2^ value, points to areas ripe for further research to optimize this property fully. Additionally, interaction terms in the compression set model draw attention to the nuanced interplay between processing parameters, underscoring the complex challenge of TPV optimization. This complexity invites a comprehensive exploration of how these parameters interactively affect TPVs, encouraging ongoing enhancements in processing techniques to improve the performance and quality of NR/EBC TPVs across diverse applications.

### 5.4. Optimization of NR/EBC TPVs Properties

The optimization process for NR/EBC TPVs relies on an expertly fine-tuned blend of natural rubber and ethylene–butene copolymer, along with specific processing parameters, to cultivate a bespoke array of material properties. As detailed in Table 12, the selected 60/40 NR/EBC blend ratio (A_4_) was subjected to an optimized processing regimen, which included a mixing temperature of 80 °C (B_1_), a rotor speed of 80 rpm (C_1_), and a mixing duration of 20 min (D_1_). This regimen was designed not only to bolster the mechanical properties of the TPVs but also to maximize their production efficiency, striking a balance between material performance and manufacturing practicality.

Table 13 thoroughly summarizes the anticipated mechanical outcomes of this optimized blend, covering key attributes such as hardness, tear strength, compression set, elongation at break, and tensile strength, alongside their statistical accuracy, as evidenced by standard errors, confidence intervals, and prediction intervals. Notably, the tensile strength of 5.39 MPa is anticipated to be strong, highlighting a considerable resistance to stress with high reliability, reflected in the tight confidence and prediction intervals. Elongation at break is expected to demonstrate significant ductility, albeit with a variability possibly inherent to the material’s testing behavior. The compression set forecast suggests a minimal tendency for permanent deformation, with a tear strength predicted to be solidly reliable and a hardness level indicating a medium firmness consistent across batches. These projections underscore the 60/40 NR/EBC blend potential to produce TPVs with the resilience and flexibility needed for diverse applications, affirming the strategic importance of this optimization process in achieving precise, reliable material specifications.

Part II: Validating the influence of mixing parameters on TPVs: morphological, rheological, and dynamic mechanical perspectives.

This part explores the significant impact of varying mixing conditions on the properties of NR/EBC TPVs, explicitly examining their phase morphology, rheological behavior, and dynamic mechanical properties. The goal is to utilize these insights to refine TPV performance by strategically adjusting mixing conditions. Table 14 details the synthesis of TPVs using cam rotors with varying blend ratios and conditions ranging from mild to severe. This methodology aims to enhance the understanding of how modifications in microstructure, achieved through processing optimization, influence the rheological and dynamic mechanical characteristics of TPVs. The study focuses on mixing conditions categorized as mild (selected optimal) and severe, emphasizing higher blend ratios (50−60%wt. of NR content), endeavoring to pinpoint the optimal processing parameters for crafting NR/EBC TPVs with the preferred properties.

### 5.5. The Relationship between Phase Morphology, Crosslink Density, and Mixing/Blending Conditions

Morphology assumes a pivotal role in governing the solid-state properties of thermoplastic vulcanizates (TPVs). The formation of a characteristic TPV-like morphology after dynamic vulcanization is imperative. The morphology engendered after this process was meticulously examined using atomic force microscopy (AFM) analysis. Figure 7 and Figure 8 depict the AFM phase images of TPVs; parts (a) and (b) capture the 50/50 and 60/40 NR/EBC blend ratios under mild (selected optimal) and severe conditions, respectively, accentuating the bright phase for NR and dark phase for EBC, coupled with the associated NR particle size distribution (a’, b’). The phase images evidence light yellow regions, indicative of crosslinked rubber particles, juxtaposed against a backdrop of dark brown areas denoting the EBC phase. It has been extensively documented that TPVs are characterized by micron-sized rubber particles uniformly dispersed within a continuous thermoplastic matrix phase [6,29]. An in-depth analysis of the data presented in Table 15, which elucidates the diameter of the rubber particles and the degree of crosslinking in NR/EBC TPVs under both mild (selected optimal) and severe conditions for 50/50 and 60/40 blend ratios, and the AFM phase images under various mixing conditions, unveils a nuanced interplay between phase morphology and crosslink density, profoundly influenced by the blend ratios and the intensity of the mixing process.

Under mild conditions, as depicted in Figure 7 for the 50/50 blend and Figure 8 for the 60/40 NR/EBC blend, there is a more uniform phase distribution, as ascertained by AFM images (a) and the narrower particle size distributions (a’). These conditions promote a more consistent distribution of NR particles throughout the EBC matrix, essential for uniform material properties. The data indicate that an increased NR content leads to higher crosslink density under mild conditions, suggesting that these conditions are conducive to comprehensive crosslinking reactions, which is particularly noticeable in the 60/40 NR/EBC blend.

Conversely, under severe mixing conditions, there is a noticeable increase in the average particle size for the 50/50 blend (see Figure 7b’) and a decrease for the 60/40 blend (see Figure 8b’). The broader particle size distribution observed under these conditions, especially with the 50/50 blend, indicates a less uniform phase morphology, which could lead to inconsistent mechanical properties. The crosslink density is slightly higher for the 50/50 blend under severe conditions compared to mild conditions, yet it shows a decrease for the 60/40 blend, as noted in Table 14. This implies that severe conditions may not consistently enhance crosslinking, and could even hinder or degrade crosslinking at higher NR contents, highlighting the intricate balance between mixing conditions, phase morphology, and crosslink density.

This analysis delineates a clear relationship between phase morphology, crosslink density, and mixing/blending conditions. Mild mixing conditions give a uniform phase distribution and stable crosslink density, likely resulting in TPVs with predictable and desirable properties. However, while severe mixing conditions might aid in reducing particle sizes, they risk introducing variability in phase morphology and crosslink density, potentially impacting material performance in an unforeseen manner.

Thermoplastic vulcanizates (TPVs) can be prepared using various mixing techniques [10]. This study, however, harnessed the power of the phase mixing method to control the TPV structure through dynamic vulcanization. The process involves premixing a crosslinker with the elastomer at a low temperature to create a curative-containing masterbatch. This masterbatch is added to the molten thermoplastic phase at a high shear rate and mixing temperature, where dynamic vulcanization and TPV formation occur. Figure 9 depicts a schematic of the development of the controlled morphology.

Dynamic vulcanization is critical in producing NR/EBC TPVs, where careful control of mixing temperature, rotor speed, and mixing time directly influences the final material properties. Under mild, optimal conditions, moderate temperatures, balanced rotor speeds, and adequate mixing times ensure the uniform dispersion of crosslinked NR particles within the EBC matrix. This yields a stable phase morphology and high crosslink density, which lead to superior mechanical properties such as enhanced tensile strength and elasticity (as discussed in Section 5.6.1). Figure 7 and Figure 8 highlight that these optimal conditions promote consistent particle size distribution, which is crucial for achieving reliable and predictable material performance. The schematic in Figure 9 further illustrates that the appropriate control of these parameters facilitates the transformation from a co-continuous phase to a stable dispersion of micron-sized vulcanized NR domains within the EBC matrix, ensuring predictable and consistent material performance.

It is important to note that severe mixing conditions, characterized by high temperatures, excessive rotor speeds, and extended mixing times, could have detrimental effects. These include partial degradation of the NR phase, irregular particle distribution, and reduced crosslink density. As a result, the mechanical properties become less predictable, and the overall material performance is reduced. This is evident in the broader particle size distributions and inconsistent morphologies, as illustrated in Figure 7b and Figure 8b. The schematic in Figure 9 further underscores the necessity of maintaining optimal mixing parameters to achieve the desired phase morphology and mechanical properties. By carefully managing these parameters, one can optimize the dynamic vulcanization process, ensuring the production of high-quality NR/EBC TPVs suitable for a wide range of industrial applications, achieving consistent performance and durability.

### 5.6. Mechanical, Rheological, and Dynamic Mechanical Properties

#### 5.6.1. Physico-Mechanical Properties

The mechanical attributes of NR/EBC TPVs are crucial for their application effectiveness and are deeply influenced by their phase morphology and crosslink density [2,22,30]. Investigations using AFM phase images and particle size analyses under varying mixing conditions indicate that mild processing conditions lead to uniform rubber dispersion, ensuring predictable and consistent mechanical properties, as detailed in Table 15. A finer particle distribution is directly linked to uniform tensile properties, which emphasizes the role of particle size uniformity in enhancing material performance [10,31,32]. High crosslink density, which increases material stiffness by restricting polymer chain movement, is particularly beneficial, as evidenced in the 60/40 NR/EBC blend processed under mild conditions, yielding higher elasticity and strength.

In essence, the analysis of the stress–strain behavior and microstructural characteristics of NR/EBC TPVs, as illustrated in Figure 10a,b (see the summary results in Table A2 in Appendix B) and supported by data in Table 15, elucidates the impact of blend ratios and processing conditions on mechanical performance. Under mild processing conditions, the 50/50 and 60/40 blends exhibit superior stress endurance up to a 500% strain threshold, a performance attributed to favorable phase morphology and enhanced crosslinking. This uniform phase distribution and increased crosslink density, especially noted in the 60/40 blends under mild conditions, resulted in a higher initial modulus and yield strength, characteristics that are crucial for applications subjected to low to moderate stress levels.

Further, the relationship between the microstructure, specifically the phase morphology, crosslink density, and particle size distribution, and the mechanical behavior of TPVs highlights the importance of controlled processing to achieve optimal material properties. The uniform dispersion and size of rubber particles contribute to an even distribution of stress and strain, improving the material’s overall tensile strength and toughness. Additionally, the stiffness and elasticity of the TPVs, which are critical for their recovery to their original shape after deformation, are significantly influenced by the crosslink density within the material.

In summary, mild processing conditions are conducive to developing TPVs with a microstructure that enhances their mechanical performance, characterized by higher stiffness, strength, and consistent material properties. This is particularly evident in blends with a higher NR content, where a more uniform phase morphology and crosslink density are critical. Conversely, severe processing conditions introduce unpredictability in mechanical properties due to a broader particle size distribution and inconsistent crosslink density, potentially detracting from the material’s performance. This underscores the critical role of processing conditions in tailoring the mechanical properties of TPVs to meet specific application requirements, ensuring optimal performance and durability.

#### 5.6.2. Viscoelastic Properties of NR/EBC TPVs

As evidenced by the strain sweep test results, TPVs commonly show linear and non-linear viscoelastic behaviors [33,34]. In the linear viscoelastic region, the responses to minor strains directly correspond to the applied stress, demonstrating the elastic and viscous responses of the materials through their elastic (G′) and loss moduli (G″). This characteristic permits reversible deformation without permanent alterations post-stress removal. At elevated strains, non-linear viscoelasticity emerges, leading to a non-linear relationship between stress and strain and the potential for irreversible deformation, emphasizing the need to differentiate these behaviors based on application-specific performance criteria.

Figure 11 indicates that the stiffness of NR/EBC TPVs, reflected by the elastic modulus (G′), is affected by the diameter of rubber particles and their crosslink density. Smaller particles suggest enhanced dispersion and greater stiffness, while a higher crosslink density limits the movement of polymer chains, further increasing stiffness [35]. Importantly, the 60/40 NR/EBC blend exhibits a higher crosslink density under mild conditions, despite its larger particle size (as outlined in Table 15), leading to increased stiffness due to tighter network interactions.

Severe processing conditions change the characteristics of the 60/40 blend, i.e., decreasing both particle size and crosslink density. This alteration impacts stiffness, demonstrating a trade-off between enhanced dispersion and diminished crosslinking. Conversely, the 50/50 blend under severe conditions leads to minor particle size and crosslink density increases, possibly maintaining stiffness levels similar to those under mild conditions.

Ultimately, the viscoelastic properties, and thus the performance, of NR/EBC TPVs, as indicated by the elastic modulus, are determined by the blend composition, particle size, and crosslink density—factors that vary with processing conditions. The 60/40 blend processed under mild conditions is ideal for applications requiring increased stiffness, which can be attributed to its significant crosslink density. Adjusting processing conditions and blend ratios enables the customization of TPV features to suit specific requirements, highlighting the crucial role of meticulous mixing in attaining the desired material attributes.

The strain dependence of the Payne effect plays a pivotal role in comprehending the viscoelastic nature of TPVs [33,36,37], particularly in formulations devoid of filler particles. Here, the interplay between polymer network dynamics and polymer–polymer interactions becomes central to understanding material behavior under strain. The absence of fillers shifts the focus to network entanglement, crosslink density, and molecular mobility as critical determinants of the material’s elasticity, viscosity, and overall mechanical response. Such dynamics are exemplified in the Payne-like effect, where variations in crosslink density markedly influence the complex shear modulus (G*), as observed in strain sweep tests. This effect illustrates the critical balance required to maintain the TPV’s structural integrity at low strains while accommodating the network’s adaptive reorganization at higher strains, directly correlating polymer network attributes with the TPV’s stiffness and resilience.

#### 5.6.3. Dynamic Mechanical Properties of TPVs

This section delves into the dynamic mechanical properties of NR/EBC TPVs, which were manipulated by varying the blend ratios and mixing conditions, including mild (selected optimal) and severe conditions. The primary objective revolves around analyzing the relationships between the dynamic mechanical properties and phase morphology of the TPVs, so that the knowledge gained could be used to enhance their mechanical and rheological properties for specific applications. This study emphasizes NR and EBC owing to their distinctive characteristics. NR contributes outstanding resilience and flexibility, while EBC augments thermoplastic processability and softens the TPV materials. Combining these materials paves the way for creating TPVs with tailored properties suitable for diverse applications. However, the interfacial interactions between the NR and EBC phases, the degree of crosslinking, and the dispersion of NR particles within the thermoplastic matrix significantly impact the viscoelastic behavior of the TPVs [19,38]. Therefore, dynamic mechanical analysis (DMA) testing facilitates the measurement of the storage modulus (E′), loss modulus (E″), loss tangent (tan δ), and other crucial parameters, providing insights into the structure–property relationships of these materials [39,40].

Figure 12 demonstrates that the viscoelastic properties of NR/EBC TPVs, such as the loss tangent (tan δ) and storage modulus (E′), are significantly influenced by the material microstructure across different temperatures and blend ratios. The summary results for NR/EBC TPVs at 50/50 and 60/40 blend ratios can be found in Table A2 in Appendix B, further highlighting the importance of our findings. The microstructural factors that impact these properties, including phase morphology, rubber particle size, and crosslink density, are illustrated in Figure 7 and Figure 8, and Table 15.

Under mild processing conditions, a more uniform phase morphology typically results in stiffer materials at lower temperatures (higher E′), indicating a well-defined glassy state and a sharper tan δ peak, suggesting efficient energy dissipation mechanisms during the glassy-to-rubbery transition. Moreover, smaller and more uniformly distributed rubber particles within the EBC matrix, achieved under mild conditions, enhance the viscoelastic performance by contributing to a higher initial E′ and affecting the position and intensity of the tan δ peak. The crosslink density, generally higher for the 60/40 blend ratio under mild conditions, directly impacts material stiffness (higher initial E′) and the temperature at the tan δ peak, as more energy is required to mobilize the crosslinked network.

The combination of uniform phase morphology, optimal particle size distribution, and appropriate crosslink density, especially under mild mixing conditions, is crucial for enhancing the viscoelastic properties of NR/EBC TPVs. This optimization results in materials exhibiting desirable mechanical behavior, such as higher stiffness and efficient energy dissipation over a broad temperature range, aligning with the requirements of various applications demanding thermal and mechanical performance.

#### 5.6.4. Glass Transition Temperature of NR/EBC TPVs

Dynamic mechanical analysis (DMA) results of NR/EBC TPVs and their base polymers uncover their thermal conduct and loss tangent (tan δ) patterns across a varied temperature range, as illustrated in Figure 13 and elaborated in Table 16. NR is highlighted for its exceptional damping abilities, with a glass transition temperature (T_g_) at −41.6 °C, showing notable flexibility at lower temperatures due to significant molecular motion. On the other hand, EBC presents a T_g_ at −32.7 °C, indicating a slightly higher temperature requirement for achieving flexibility.

There is a tan δ variation from 50 °C to 80 °C, identifying a phase transition between 55 and 60 °C that corresponds with the EBC’s melting point (T_m_ = 62 °C). At this transition, EBC adopts a rubbery state, boosting its damping capability. However, as the temperature further increases, tan δ decreases, suggesting a shift toward a more fluid-like state with reduced viscoelastic properties, largely attributed to the NR’s vulcanized network [19]. A single broad damping peak in TPVs complicates the straightforward T_g_ identification of each polymer. Nonetheless, through deconvolution techniques employing Gaussian or Lorentzian functions [39,40], these broad peaks have been effectively separated, revealing the effects of different NR and EBC content ratios on T_g_ and damping behaviors in TPVs, with an increased NR content leading to more noticeable damping peaks under various processing conditions.

The temperature-dependent loss tangent analysis and its deconvolution underscore the crucial impact of TPV microstructure on viscoelastic properties. Uniform phase morphology, achieved under mild processing conditions, yields a sharp and distinct tan δ peak, indicating an efficient energy dissipation and a well-defined T_g_. This affirms that a uniformly dispersed rubber phase favorably affects mechanical damping properties. The viscoelastic response’s dependency on the rubber particles’ diameter and distribution is revealed in Table 15, where a more consistent particle size distribution under mild conditions produces sharper tan δ peaks, suggesting that particle uniformity promotes damping behavior by maintaining consistent molecular mobility.

Furthermore, the degree of crosslinking, as noted in Table 15, significantly influences T_g_, as observed in the deconvoluted tan δ curves (Figure 13b). Figure A1 in Appendix B illustrates the total number of observations in the TPV’s deconvoluted tan δ curves. Increased crosslink density under mild conditions heightens the T_g_ of the NR phase, indicating that crosslinks constrain polymer chain movement and elevate the temperature necessary for the glass-to-rubber transition. The broader tan δ peaks in TPVs, compared to neat NR and EBC, reflect the blend complexity influenced by each polymeric phase. Deconvolution of these overlapping transitions unveils the intricate NR-EBC interactions within the TPV microstructure, providing deep insights into how factors like phase morphology, particle size and distribution, and crosslink density dictate TPVs’ viscoelastic behavior. This underscores the importance of fine-tuning these microstructural elements to enhance performance, especially in applications where the mechanical properties’ temperature sensitivity is critical.

## 6. Conclusions

Utilizing a comprehensive full factorial design of experiments (DOE) approach, we successfully identified the optimal processing conditions for creating thermoplastic vulcanizates (TPVs) from natural rubber (NR) and ethylene–butene copolymer (EBC). Vital mechanical properties such as tensile strength, elongation at break, compression set, tear strength, and hardness were meticulously analyzed, revealing the significant impacts of variables, including blend ratio, mixing temperature, mixing time, and rotor speed. Through analysis of variance (ANOVA), the research pinpointed essential processing parameters and their interplay, establishing a set of optimal conditions that significantly improved the TPVs’ performance. The selected 60/40 NR/EBC blend ratio, processed at 80 °C for 20 min at 80 rpm, demonstrated enhanced mechanical attributes and optimized production efficiency, balancing high-quality material performance with practical manufacturing concerns.

The predictive model guidance in fine-tuning NR/EBC blends represents a significant advancement in TPV development. Validation through atomic force microscopy (AFM) confirmed the success of the dynamic vulcanization process. The rubber process analyzer (RPA) and dynamic mechanical analyzer (DMA) assessments aligned well with the optimized parameters, demonstrating effective control over the TPV’s microstructure and behavior. Deconvolution analysis of the glass transition temperature (T_g_) from tan δ peaks provided detailed insights into microstructural dynamics, further refining our knowledge of TPV performance for specific applications through a nuanced understanding of crosslink density and phase morphology. The enhanced properties of NR/EBC TPVs demonstrated in the study make them highly suitable for various industrial applications, including automotive, consumer goods, construction, electrical, healthcare, and packaging. Combining the elasticity of natural rubber with the toughness of EBC offers a versatile, durable, and environmentally friendly material, ideal for producing durable seals, gaskets, hoses, footwear components, weatherproofing materials, cable insulation, medical tubing, and flexible packaging.

The study successfully identified the optimal processing conditions for NR/EBC TPVs and demonstrated their enhanced mechanical properties. However, it has some limitations, such as having been conducted on a laboratory scale, needing more long-term performance data, and the need to address material variability from different suppliers. Additionally, recyclability, comprehensive thermal stability, aging resistance, and biodegradability analyses were not included. Future research should focus on industrial-scale validation, long-term performance studies, material sourcing variability, broader property analysis, application-specific research, and advanced characterization techniques. Addressing these aspects will further optimize NR/EBC TPVs, enhancing their industrial applicability and performance.

## Figures and Tables

**Figure 1 polymers-16-01963-f001:**
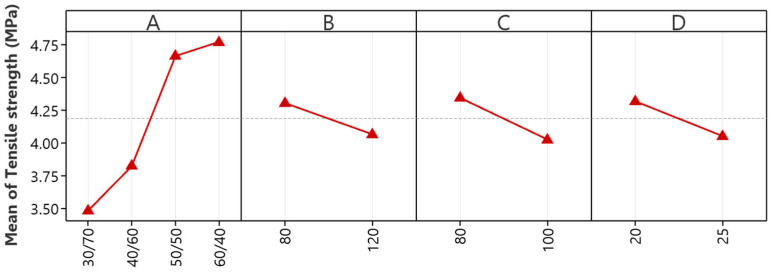
The main effects plots for tensile strength: NR/EBC blend ratio (A); mixing temperature (B); rotor speed (C); and mixing time (D).

**Figure 2 polymers-16-01963-f002:**
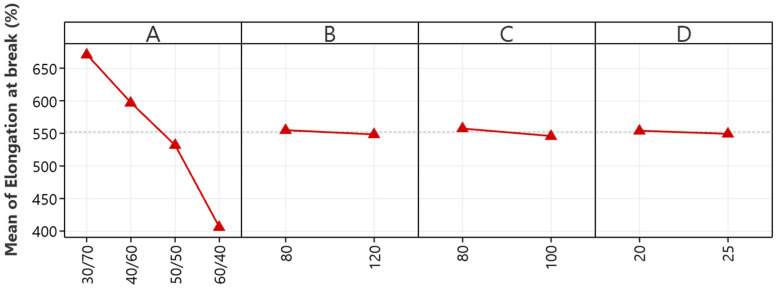
The main effects plots for elongation at break: NR/EBC blend ratio (A); mixing temperature (B); rotor speed (C); and mixing time (D).

**Figure 3 polymers-16-01963-f003:**
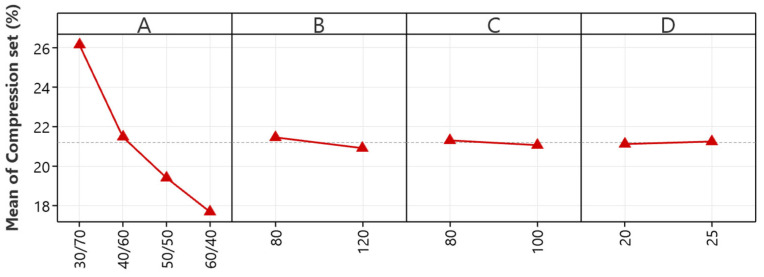
The main effects plots for compression set: NR/EBC blend ratio (A); mixing temperature (B); rotor speed (C); and mixing time (D).

**Figure 4 polymers-16-01963-f004:**
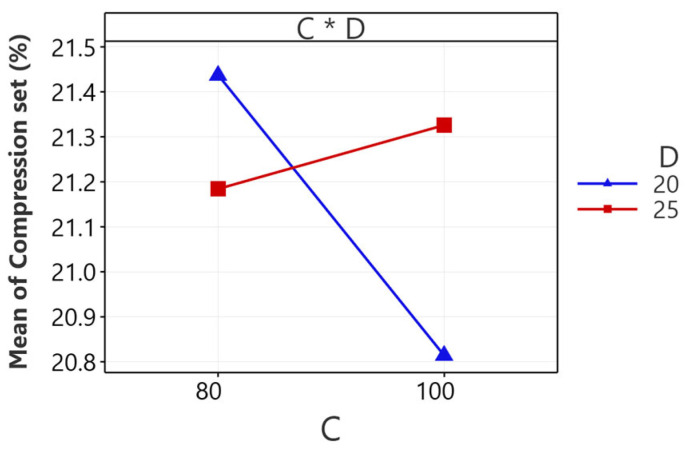
The interaction effect plot for compression set: NR/EBC blend ratio (A); mixing temperature (B); rotor speed (C); and mixing time (D).

**Figure 5 polymers-16-01963-f005:**
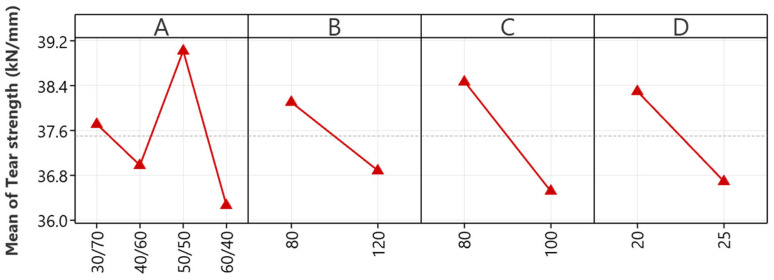
The main effects plots for tear strength: NR/EBC blend ratio (A); mixing temperature (B); rotor speed (C); and mixing time (D).

**Figure 6 polymers-16-01963-f006:**
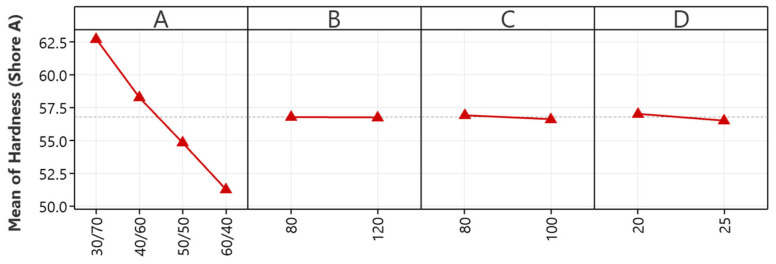
The main effect plots for hardness: NR/EBC blend ratio (A); mixing temperature (B); rotor speed (C); and mixing time (D).

**Figure 7 polymers-16-01963-f007:**
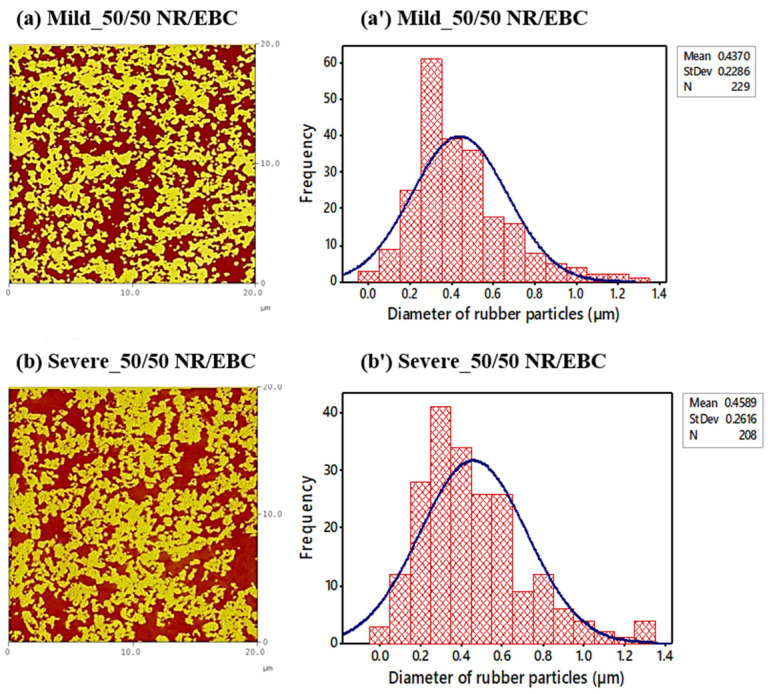
AFM phase images of TPVs: (**a**,**b**) using cam rotors under mild (selected optimal) and severe conditions for 50/50 NR/EBC blend ratios (bright phase: NR; dark phase: EBC), alongside associated NR particle size distribution (**a’**,**b’**).

**Figure 8 polymers-16-01963-f008:**
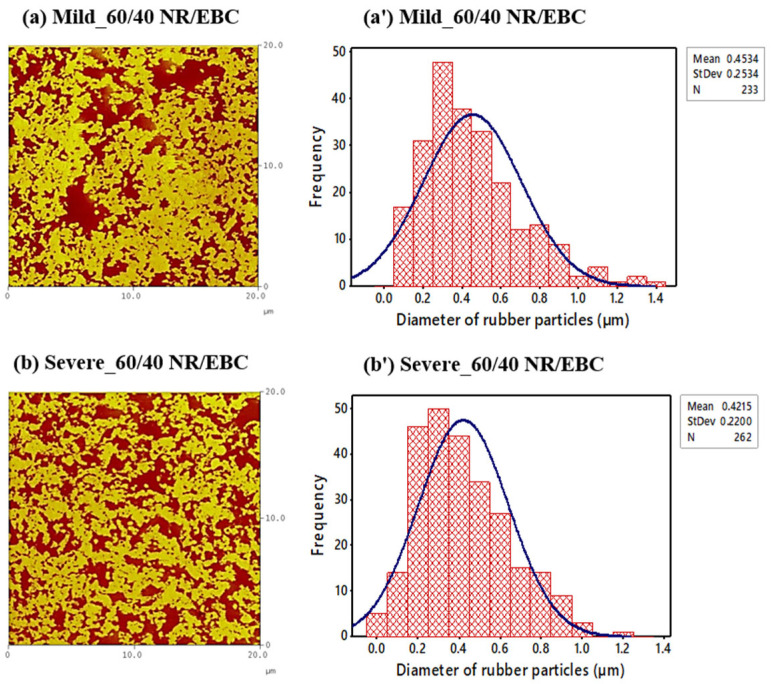
AFM phase images of TPVs: (**a**,**b**) using cam rotors under mild (selected optimal) and severe conditions for 60/40 NR/EBC blend ratios (bright phase: NR; dark phase: EBC), alongside associated NR particle size distribution (**a’**,**b’**).

**Figure 9 polymers-16-01963-f009:**
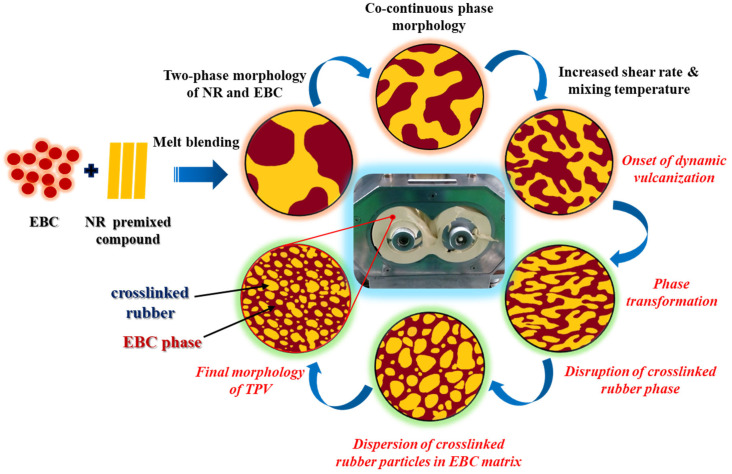
Morphology evolution of TPVs during dynamic vulcanization based on NR/EBC blend.

**Figure 10 polymers-16-01963-f010:**
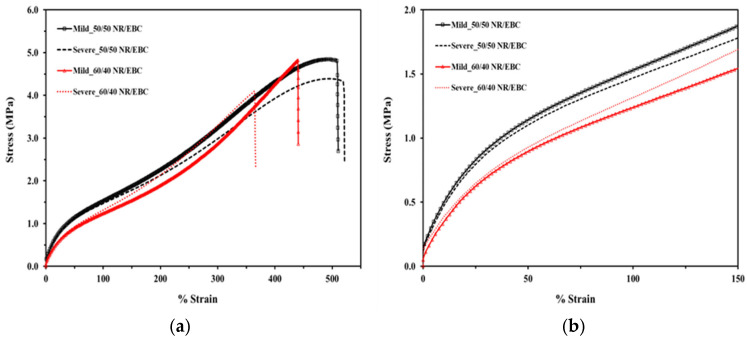
Stress–strain curves for NR/EBC TPVs at the 50/50 and 60/40 NR/EBC blends under mild (selected optimal) and severe processing conditions across various blend ratios: (**a**) comprehensive overview; (**b**) focused on 0–150% strain range.

**Figure 11 polymers-16-01963-f011:**
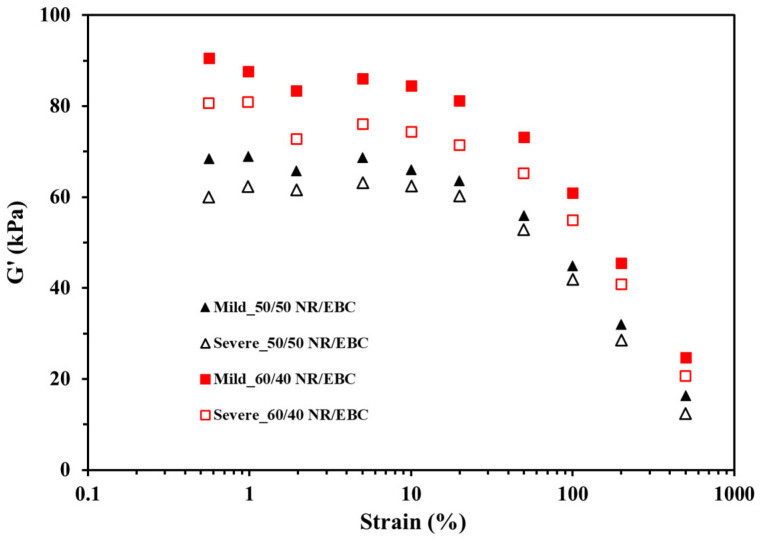
Elastic modulus (G′) as a function of strain (Payne effect) of NR/EBC TPVs prepared under mild (selected optimal) and severe conditions for 50/50 and 60/40 blend ratios.

**Figure 12 polymers-16-01963-f012:**
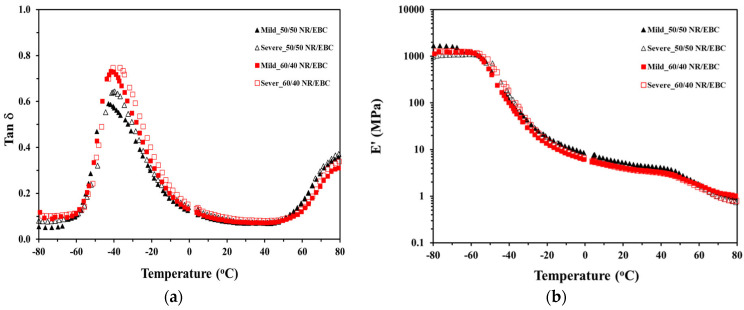
Temperature-dependent behaviors: (**a**) loss tangent (tan δ) and (**b**) storage modulus (E′) in NR/EBC TPVs, prepared under mild (selected optimal) and severe mixing conditions for 50/50 and 60/40 blend ratios.

**Figure 13 polymers-16-01963-f013:**
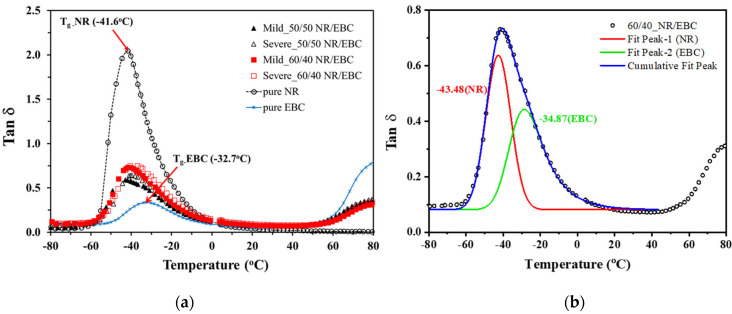
Temperature-dependent loss tangent (tan δ): (**a**) for parent polymers (NR, EBC) and NR/EBC TPVs; and (**b**) illustrating the deconvolution method applied to the tan δ peak for TPVs.

**Table 1 polymers-16-01963-t001:** Compounding recipe of NR premixed compound.

Materials	Amount (phr)
Natural rubber (STR 5L)	100
Zinc oxide (ZnO)	5
Stearic acid	1
Phenolic resin (SP1045)	10
Stannous chloride (SnCl_2_)	1
Wingstay L	1

**Table 2 polymers-16-01963-t002:** Mixing sequence for preparing NR premixed compound.

Time (min)	Action
0	Add NR
3	Add stearic acid and ZnO
5	Add phenolic resin and SnCl_2_
10	Add Wingstay L
12	Discharge

**Table 3 polymers-16-01963-t003:** TPV composition with various blend ratios between NR premixed compound and EBC.

NR/EBC Blend Ratios	NR Premixed Compound (%wt)	EBC (%wt)
30/70	30	70
40/60	40	60
50/50	50	50
60/40	60	40

**Table 4 polymers-16-01963-t004:** Selected processing parameters and their respective levels.

Parameter	Mixing Conditions
Rotor geometry type	Cam rotor
Fill factor	0.75
NR/EBC blend ratios (by weight)	30/70, 40/60, 50/50, 60/40
Initial mixing temperature (°C)	80, 120
Rotor speed (rpm)	80, 100
Mixing time (min)	20, 25

**Table 5 polymers-16-01963-t005:** Mixing conditions for preparing NR/ECB TPVs using an internal mixer.

Parameter Variables	Units	Factors	Levels			
			1	2	3	4
NR/EBC Blend Ratio	%wt	A	30/70	40/60	50/50	60/40
Mixing Temperature	°C	B	80	120	-	-
Rotor Speed	rpm	C	80	100	-	-
Mixing Time	min	D	20	25	-	-

**Table 6 polymers-16-01963-t006:** Analysis of variance results for tensile strength of NR/EBC TPVs.

Source	DF	SS	MS	*F*-Value	*p*-Value
Model	18	23.4103	1.30057	15.97	0.000 *
A	3	19.1977	6.39922	78.59	0.000 *
B	1	0.9084	0.90845	11.16	0.002 *
C	1	1.6272	1.62722	19.98	0.000 *
D	1	1.1329	1.13289	13.91	0.001 *
AB	3	0.0878	0.02926	0.36	0.783
AC	3	0.0530	0.01766	0.22	0.884
AD	3	0.3238	0.10795	1.33	0.278
BC	1	0.0686	0.06858	0.84	0.364
BD	1	0.0106	0.01063	0.13	0.719
CD	1	0.0002	0.00024	0.00	0.957
Error	45	3.6641	0.08142		
Lack of Fit	13	0.2214	0.01703	0.16	0.999
Pure Error	32	3.4427	0.10758		
Total	63	27.0744			
Model Summary: S = 0.285349; R^2^ = 86.47%; R^2^(adj) = 81.05%; R^2^(pred) = 72.63%					

* *p*-Value < 0.05 is significant.

**Table 7 polymers-16-01963-t007:** Analysis of variance for the elongation at break of NR/EBC TPVs.

Source	DF	SS	MS	*F*-Value	*p*-Value
Model	18	616,839	34,269	33.64	0.000 *
A	3	608,833	202,944	199.24	0.000 *
B	1	683	683	0.67	0.417
C	1	2093	2093	2.05	0.159
D	1	365	365	0.36	0.552
AB	3	1613	538	0.53	0.665
AC	3	2055	685	0.67	0.573
AD	3	522	174	0.17	0.916
BC	1	0	0	0.00	0.993
BD	1	55	55	0.05	0.817
CD	1	619	619	0.61	0.440
Error	45	45,836	1019		
Lack of Fit	13	3839	295	0.23	0.997
Pure Error	32	41,997	1312		
Total	63	662,675			
Model Summary: S = 31.9152; R^2^ = 93.08%; R^2^(adj) = 90.32%; R^2^(pred) = 86.01%					

* *p*-Value < 0.05 is significant.

**Table 8 polymers-16-01963-t008:** Analysis of variance for the compression set of NR/EBC TPVs.

Source	DF	SS	MS	*F*-Value	*p*-Value
Model	18	658.693	36.594	68.65	0.000 *
A	3	645.845	215.282	403.86	0.000 *
B	1	4.759	4.759	8.93	0.005 *
C	1	0.930	0.930	1.74	0.193
D	1	0.268	0.268	0.50	0.482
AB	3	2.454	0.818	1.53	0.219
AC	3	1.385	0.462	0.87	0.466
AD	3	0.215	0.072	0.13	0.939
BC	1	0.067	0.067	0.13	0.724
BD	1	0.424	0.424	0.80	0.377
CD	1	2.346	2.346	4.40	0.042 *
Error	45	23.988	0.533		
Lack of Fit	13	3.342	0.257	0.40	0.960
Pure Error	32	20.645	0.645		
Total	63	682.681			
Model Summary: S = 0.730107; R^2^ = 96.49%; R^2^(adj) = 95.08%; R^2^(pred) = 92.89%					

* *p*-Value < 0.05 is significant.

**Table 9 polymers-16-01963-t009:** Analysis of variance for the tear strength of NR/EBC TPVs.

Source	DF	SS	MS	*F*-Value	*p*-Value
Model	18	227.893	12.6607	3.42	0.000 *
A	3	66.783	22.2610	6.01	0.002 *
B	1	23.766	23.7656	6.42	0.015 *
C	1	61.623	61.6225	16.65	0.000 *
D	1	41.281	41.2806	11.15	0.002 *
AB	3	10.888	3.6294	0.98	0.410
AC	3	11.014	3.6713	0.99	0.405
AD	3	1.721	0.5735	0.15	0.926
BC	1	9.302	9.3025	2.51	0.120
BD	1	1.266	1.2656	0.34	0.562
CD	1	0.250	0.2500	0.07	0.796
Error	45	166.567	3.7015		
Lack of Fit	13	14.997	1.1536	0.24	0.995
Pure Error	32	151.570	4.7366		
Total	63	394.459			
Model Summary: S = 1.92392; R^2^ = 57.77%; R^2^(adj) = 40.88%; R^2^(pred) = 14.59%					

* *p*-Value < 0.05 is significant.

**Table 10 polymers-16-01963-t010:** Analysis of variance for the hardness (Shore A) of NR/EBC TPVs.

Source	DF	SS	MS	*F*-Value	*p*-Value
Model	18	1158.01	64.334	147.48	0.000 *
A	3	1148.32	382.774	877.46	0.000 *
B	1	0.01	0.008	0.02	0.892
C	1	1.42	1.416	3.25	0.078
D	1	4.26	4.264	9.78	0.003 *
AB	3	2.17	0.723	1.66	0.190
AC	3	0.98	0.326	0.75	0.530
AD	3	0.31	0.105	0.24	0.868
BC	1	0.24	0.235	0.54	0.467
BD	1	0.15	0.152	0.35	0.558
CD	1	0.15	0.152	0.35	0.558
Error	45	19.63	0.436		
Lack of Fit	13	2.13	0.164	0.30	0.988
Pure Error	32	17.50	0.547		
Total	63	1177.64			
Model Summary: S = 0.660478; R^2^ = 98.33%; R^2^(adj) = 97.67%; R^2^(pred) = 96.63%					

* *p*-Value < 0.05 is significant.

**Table 11 polymers-16-01963-t011:** Regression model summary for response variables after using reduced model.

Response	Regression Equation	R^2^	R^2^(adj)	R^2^(pred)
Tensile strength (MPa)	Y = 4.1868 − 0.7034A_1_ − 0.3624A_2_ + 0.4788A_3_ + 0.5870A_4_ + 0.1191B_1_ − 0.1191B_2_ + 0.1595C_1_ − 0.1595C_2_ + 0.1330D_1_ − 0.1330D_2_	84.46%	82.82%	80.41%
Elongation at break (%)	Y = 551.87 + 119.84A_1_ + 45.61A_2_ − 19.77A_3_ − 145.67A_4_ + 3.27B_1_ − 3.27B_2_ + 5.72C_1_ − 5.72C_2_ + 2.39D_1_ − 2.39D_2_	92.35%	91.54%	90.35%
Compression set (%)	Y = 21.1903 + 4.985A_1_ + 0.293A_2_ − 1.775A_3_ − 3.503A_4_ + 0.2727B_1_ − 0.2727B_2_ + 0.1206C_1_ − 0.1206C_2_ − 0.0647D_1_ + 0.0647D_2_ + 0.1915(CD)_11_ − 0.1915(CD)_12_ − 0.1915(CD)_21_ + 0.1915(CD)_22_	95.82%	95.30%	94.54%
Tear strength (kN/mm)	Y = 37.497 + 0.222A_1_ − 0.516A_2_ + 1.528A_3_ − 1.234A_4_ + 0.609B_1_ − 0.609B_2_ + 0.981C_1_ − 0.981C_2_ + 0.803D_1_ − 0.803D_2_ + 0.381(BC)_11_ − 0.381(BC)_12_ − 0.381(BC)_21_ + 0.381(BC)_22_	51.40%	45.33%	36.52%
Hardness (Shore A)	Y = 56.7794 + 5.957A_1_ + 1.496A_2_ − 1.954A_3_ − 5.498A_4_ + 0.0113B_1_ − 0.0113B_2_ + 0.1488C_1_ − 0.1488C_2_ + 0.2581D_1_ − 0.2581D_2_	97.99%	97.78%	97.47%

**Table 12 polymers-16-01963-t012:** The optimal mixing conditions for NR/EBC TPVs.

Variable Parameters	Code Level	Condition Setting
NR/EBC blend ratio	A_4_	60/40
Mixing temperature (°C)	B_1_	80
Rotor speed (rpm)	C_1_	80
Mixing time (min)	D_1_	20

**Table 13 polymers-16-01963-t013:** The response predictions for the NR/EBC TPVs obtained from optimal conditions.

Response	Target	Fit	SE Fit	95%CI	95%PI
Tensile strength (MPa)	max.	5.39	0.155	(5.081, 5.707)	(4.739, 6.048)
Elongation at break (%)	min.	406.2	7.490	(391.22, 421.18)	(344.43, 467.96)
Compression set (%)	min.	18.21	0.252	(17.701, 18.712)	(16.690, 19.723)
Tear strength (kN/mm)	max.	39.04	0.654	(37.727, 40.348)	(35.106, 42.969)
Hardness (Shore A)	min.	51.7	0.213	(51.273, 52.126)	(50.341, 53.057)

**Table 14 polymers-16-01963-t014:** The mixing conditions of TPVs under mild (selected optimal) and severe mixing conditions.

Mixing Conditions	Parameter Setting		
	Mixing Temperature (°C)	Rotor Speed (rpm)	Mixing Time (min)
Mild (selected optimal)	80	80	20
Severe	120	100	25

**Table 15 polymers-16-01963-t015:** Diameter of the rubber particles and degree of crosslinking of NR/EBC TPVs under mild (selected optimal) and severe conditions at 50/50 and 60/40 blend ratios.

Mixing Conditions	NR/EBC Blend Ratios	Diameter (µm)		Crosslink Density (1 × 10^−5^ mol/cm^3^)
		Mean	SD	
Mild (selected optimal)	50/50	0.4370	0.2286	7.68 ± 0.03
	60/40	0.4534	0.2534	9.83 ± 0.30
Severe	50/50	0.4589	0.2616	7.83 ± 0.05
	60/40	0.4215	0.2200	9.03 ± 0.39

**Table 16 polymers-16-01963-t016:** Deconvolution for glass transition temperature (T_g_) analysis.

Mixing Conditions	NR/EBC Blend Ratios	T_g_ of TPVs (°C)	Mixing Conditions	
			T_g_-NR Phase (°C)	T_g_-EBC Phase (°C)
NR	100/0	−40.60	-	-
EBC	0/100	−32.70	-	-
Mild condition (selected optimal)	50/50	−43.00	−45.23	−37.21
	60/40	−41.70	−43.48	−34.87
Severe condition	50/50	−38.10	−41.30	−34.79
	60/40	−40.60	−42.39	−33.28

## Data Availability

The data presented in this study are available upon request from the corresponding author due to privacy concerns.

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
