# Peer review of "Optimizing Processing Parameters for NR/EBC Thermoplastic Vulcanizates: A Comprehensive Full Factorial Design of Experiments (DOE) Strategy"

_polymers, 2024, doi:10.3390/polym16141963_

Round 1
Reviewer 1 Report
Comments and Suggestions for Authors
In this manuscript, the authors present a study on optimizing processing parameters for thermoplastic vulcanizate (TPV) blends derived from natural rubber (NR) and ethylene-butene copolymer (EBC) using a specific blend ratio and melt mixing technique. The research employs a comprehensive full factorial design of experiments (DOE) methodology to enhance the mechanical performance of the TPVs through dynamic vulcanization. Overall, this manuscript presents a well-structured study on optimizing processing parameters for NR/EBC TPVs. With some minor revisions and additional information, the paper has the potential to contribute significantly to the field of polymer science.
1. It is suggested to provide more detailed information on the specific blend ratio used in the study for enhancing the clarity. Also, a relevant digital image of the sample should be provided.
2. The authors should elaborate on the role of dynamic vulcanization in the production process for better understanding.
3. For clearer understanding, it would be better if discussed the potential industrial applications of NR/EBC TPVs based on the study findings.
4. The use of a DOE methodology provides a systematic approach to parameter optimization, leading to achieving the enhanced mechanical damping properties and efficient energy dissipation. Likewise, an introduction of three-dimensional structures to the material system can also be effective and some efforts should be cited for more comprehensive understanding [Appl. Phys. Rev. 9, 011322 (2022) and Sci. Rep. 6, 31067 (2016)]
5. A brief mention about the study's limitations and propose future research directions can be considered to expand on the current work.
Author Response
Comment 1: It is suggested to provide more detailed information on the specific blend ratio used in the study for enhancing the clarity. Also, a relevant digital image of the sample should be provided.
Response: We have added detailed information on the preparation steps for NR/EBC TPVs on pages 3 and 4, lines 127-138. The specific proportions for mixing the NR/EBC blend are now clearly presented in Table 3 on page 4.
Comment 2: The authors should elaborate on the role of dynamic vulcanization in the production process for better understanding.
Response: We have provided an additional explanation on page 19, lines 631-637, detailing the role of dynamic vulcanization. Also, we have included a comprehensive image (Figure 9 on page 20) illustrating the mechanism of phase development during dynamic vulcanization of the NR/EBC blend.
Comment 3: For clearer understanding, it would be better if discussed the potential industrial applications of NR/EBC TPVs based on the study findings.
Response: We have expanded the Conclusion section on pages 25-26, lines 855-861, to include a discussion of potential industrial applications of NR/EBC TPVs based on our study findings.
Comment 4: The use of a DOE methodology provides a systematic approach to parameter optimization, leading to achieving the enhanced mechanical damping properties and efficient energy dissipation. Likewise, an introduction of three-dimensional structures to the material system can also be effective and some efforts should be cited for more comprehensive understanding.
Response: Thank you very much for your suggestion. Unfortunately, the details of the references are beyond the manuscript scope.
Comment 5: A brief mention about the study's limitations and propose future research directions can be considered to expand on the current work.
Response: As kindly suggested, we have included a discussion of the study limitations, and proposed future research directions in the Conclusion section on page 26, lines 862-871. This addition provides a broader context for our work and suggests avenues for further investigation.

Reviewer 2 Report
Comments and Suggestions for Authors
Review report
I have gone through the manuscript titled “Optimizing processing parameters for NR/EBC thermoplastic vulcanizates: a comprehensive full factorial design of experiments (DOE) strategy” by Phupewkeaw et al.
The authors present a factorial DOE and ANOVA analysis for optimizing the processing parameters for natural rubber/EBC thermoplastic vulcanizates. Further they present the experimental data (AFM-based morphology, stress strain data, glass transition temperature and rheological properties) of the prepared materails. The data presented is of good quality and the manuscript is systematically written. The presented work is of interest to the polymer community. However, the following points are to be addressed before the manuscript can be considered further.
1) Page no. 2, line 54-56: a few more referenced are required.
2) Page 3, line 100: please change “t o” to “to”.
3) Page 3, line 126: whether, NR was distributed as particles? I don’t think particle is the correct word.
4) What is the difference between section 3.1 and 3.6? Why a same equipment cannot be used?
5) Page 7, line 263: where is the particle dispersion? Does the authors mean natural rubber?
6) Though the authors talk elaborately on the DOE and ANOVA, it is surprising that the experimental data (like Figs. 1-5) are without error bars. Kindly address.
7) The estimated properties, obtained using the regression equations, are to be experimentally validated. Kindly comment.
8) It is not clear how morphological analyses confirmed the validity of the prediction model.
9) The stress-strain data and rheological properties are measured for the optimal mixture condition? Though data is shown for the sever/mild cases, a discussion on the underlying reasons for the observed variation is missing. Kindly check.
10) The graph and discussion on tan (delta) shall be moved before the discussion on glass transition temperatures.
11) Though there are many tables in the manuscript, a separate table highlighting the experimental data (elongation, UTS, glass transition temperature, rheological properties, etc.) are necessary.
12) One important experimental aspect is the adhesion force of the polymers, and its variation with room temperature vulcanized polymers (see: J. Elastomers & Plastics, 54, 2022, 1172-1201, DOI: 10.1177/00952443221133237). Do the authors have any data on adhesion values of the prepared materials?

Author Response
Comment 1: Page no. 2, line 54-56: a few more references are required.
Response: As kindly suggested, we have included additional references - Ref. no. [4, 7-9] on page 2, line 54.
Comment 2: Page 3, line 100: please change "t o" to "to".
Response: Thank you very much for pointing that out. We have corrected it, as suggested.
Comment 3: Page 3, line 126: whether NR was distributed as particles? I don't think particle is the correct word.
Response: Thank you for your comment. The authors have revised and explained more as shown on page 3, lines 118-138.
Comment 4: What is the difference between section 3.1 and 3.6? Why a same equipment cannot be used?
Response: The differences between sections 3.1 and 3.6 are as follows:
Section 3.1 focuses on tensile and tear properties under non-cyclic loading using a universal testing machine. The tensile and tear tests are destructive tests, i.e., the rupture of the test specimen is resulted.
Section 3.6 examines dynamic mechanical properties under sinusoidal loading using a dynamic mechanical analyzer (DMA). The DMA test is typically known as a non-destructive test, where the elastic and viscous responses are measured simultaneously.
Comment 5: Page 7, line 263: where is the particle dispersion? Does the author mean natural rubber?
Response: The authors apologize for the unclear statement. The phrase “particle dispersion” in line 263 refer to the natural rubber phase which undergoes crosslinking and forms finely dispersed rubber particles during dynamic vulcanization.
Comment 6: Though the authors talk elaborately on the DOE and ANOVA, it is surprising that the experimental data (like Figs. 1-5) are without error bars. Kindly address.
Response: Thank you for your comment. The plots as shown in Figures 1-5 are analyzed from the ANOVA, and the main effect and interaction effect plots are of interest. In other words, these plots illustrate mean effects rather than data variability. The statistical significance is typically indicated by F-values and P-values from the ANOVA. Including the error bars would clutter the visual representation.
Comment 7: The estimated properties, obtained using the regression equations, are to be experimentally validated. Kindly comment.
Response: We have explained the robustness of our regression equations, citing high R² values and insignificant Lack-of-Fit. We acknowledge the importance of experimental validation and have described our approach to ensure both statistical soundness and practical applicability.
Comment 8: It is not clear how morphological analyses confirmed the validity of the prediction model.
Response: To clarify further the validity of the prediction model via morphological analyses, the authors have provided additional explanation on pages 19-20, lines 631-660, along with a schematic illustration in Figure 9.
Comment 9: The stress-strain data and rheological properties are measured for the optimal mixture condition? Though data is shown for the severe/mild cases, a discussion on the underlying reasons for the observed variation is missing. Kindly check.
Response: As kindly pointed out, we have clarified that measurements were taken under optimal mixing conditions to establish a benchmark. While data from severe and mild conditions are included, the optimal conditions provide a reference point. This approach balances material performance with production efficiency.
Comment 10: The graph and discussion on tan(delta) shall be moved before the discussion on glass transition temperatures.
Response: As kingly suggested, we did rearrange the graph and discussion to page 23.
Comment 11: Though there are many tables in the manuscript, a separate table highlighting the experimental data (elongation, UTS, glass transition temperature, rheological properties, etc.) are necessary.
Response: We have included a summary table (Table B1 in Appendix B) with key experimental data, including tensile strength (TS), elongation at break (EB), storage modulus in the glassy region (E'g), tanδ peak, and glass temperature (Tg). Also, we have put citations in the relevant sections of the manuscript on pages 20, line 676, and 23, line 767.
Comment 12: One important experimental aspect is the adhesion force of the polymers, and its variation with room temperature vulcanized polymers (see: J. Elastomers & Plastics, 54, 2022, 1172-1201, DOI: 10.1177/00952443221133237). Do the authors have any data on adhesion values of the prepared materials?
Response: The authors deeply agree with the reviewer’s comment that the adhesion force is one of the important factors. Unfortunately, this is beyond the scope of this manuscript. However, we will consider including this adhesion force in our further research.

Round 2
Reviewer 2 Report
Comments and Suggestions for Authors
The authors have provided suitable response to the review comments and have revised the manuscript accordingly. Hence, I am happy to recommend the manuscript for publication.